# Convection Enhanced Delivery of Topotecan for Gliomas: A Single-Center Experience

**DOI:** 10.3390/pharmaceutics13010039

**Published:** 2020-12-30

**Authors:** Pavan S. Upadhyayula, Eleonora F. Spinazzi, Michael G. Argenziano, Peter Canoll, Jeffrey N. Bruce

**Affiliations:** 1Department of Neurological Surgery, Columbia University Irving Medical Center, New York, NY 10032, USA; efs2110@cumc.columbia.edu (E.F.S.); mga2122@cumc.columbia.edu (M.G.A.); jnb2@cumc.columbia.edu (J.N.B.); 2Department of Cell Biology and Pathology, Columbia University Irving Medical Center, New York, NY 10032, USA; pc561@cumc.columbia.edu

**Keywords:** convection enhanced delivery, glioma, glioblastoma, topotecan

## Abstract

A key limitation to glioma treatment involves the blood brain barrier (BBB). Convection enhanced delivery (CED) is a technique that uses a catheter placed directly into the brain parenchyma to infuse treatments using a pressure gradient. In this manuscript, we describe the physical principles behind CED along with the common pitfalls and methods for optimizing convection. Finally, we highlight our institutional experience using topotecan CED for the treatment of malignant glioma.

## 1. Introduction

Glioblastoma (GBM) is the most common and aggressive primary brain tumor, with a remarkable propensity to infiltrate the surrounding brain tissue, making complete resection impossible, recurrence inevitable, and prognosis dismal [1]. The genetic, transcriptional and functional heterogeneity of this malignancy and its microenvironment present a major challenge to the successful application of targeted therapies [2]. The current standard of care consisting of resection, temozolomide [3], and radiation [4] epitomizes the scarcity of major breakthroughs in the past 50 years and the desperate need for more comprehensive and innovative approaches to make transformative contributions in the clinical management of GBM. Although numerous chemotherapeutic drugs demonstrate a significant anti-tumor activity in preclinical studies, their efficacy in clinical trials has been disappointing since systemic delivery fails to achieve therapeutic drug levels in tumor cells due to blood-brain barrier permeability and systemic toxicity [5,6]. Therefore, circumventing the BBB remains a major research topic within CNS oncology, as it is essential for effective anti-tumor response. Multiple techniques including the use of liposomes or chemotherapy laden nanoparticles, focused ultrasound, and even surgical grafts receiving extracranial vessel blood supply are under investigation as mechanisms to bypass the BBB [7,8,9]. Convection enhanced delivery (CED), first described by Bobo et al. in 1994 [10], remains a promising technique for circumventing the BBB and delivering therapy in a non-diffusion dependent manner, thereby facilitating high local concentrations of infusate. This mechanism of bypassing the BBB dramatically increases the number of therapies that can feasibly be used to treat glioma.

Herein, we will explore the applications of CED to the glioma treatment. We will highlight our institutional experience with CED with a special focus on the topoisomerase inhibitor topotecan.

## 2. Blood Brain Barrier Physiology and Pathophysiology in Glioma

The BBB consists of a network of cells bound together by tight junctions. These cells, endothelial cells surrounding cerebral microvasculature and epithelial cells in the meningeal arachnoid and choroid plexus, work to exclude large hydrophilic molecules, thereby tightly regulating the transit of substances into the CNS. While small lipophilic molecules or gases can pass paracellularly, all other substances must be actively transported transcellularly due to the occlusive tight junctions between cells [11]. Therefore, most chemotherapeutics are excluded from the CNS due to their molecular size or polarity [12].

High-grade glioma is capable of disrupting the neurovascular unit that comprises endothelial BBB. Not only is high-grade glioma characterized by endothelial proliferation, and neovascularization, but tumor cells can physically disrupt the BBB integrity and secrete factors leading to the increased BBB permeability [13,14]. This disruption can lead to a myriad of pathologies including the loss of ionic regulation and epileptogenesis [15] or edema and neurological compromise. The areas with disrupted BBB are surgically resected when possible. The near universal recurrence of GBM highlights the importance of treatments targeting tumor cells invading peritumoral areas, where the BBB remains intact.

## 3. Convection Enhanced Delivery—Rationale and Mechanisms

Convection relies on a constant hydraulic force that distributes infusate based on a pressure gradient causing bulk flow through the interstitial space. Utilizing this pressure gradient gives CED two key benefits: A larger volume of distribution and a constant concentration of infusate within the volume of distribution. This is in contrast to non-convective methods with systemic delivery, which are diffusion-dependent and rely on a passive concentration gradient for infusate distribution. This passive flow limits infusate distribution to just a few millimeters from the drug source and requires steep concentration gradients for adequate treatment [16].

### 3.1. Biophysical Properties

CED is achieved using a catheter attached to a hydraulic pump. The catheter is inserted directly into the interstitium of the brain. Infusate is distributed through pressure equalization of the interstitial fluid. Although the different tissue densities of gray or white matter can alter the volume of distribution, the infusate is not bound by these anatomical boundaries. However, the volume of distribution is bound by the pial surface. Unlike non-convective methods where a gradual concentration gradient will occur, infusate delivery is relatively constant across the volume of distribution with a steep drop off outside this volume, which then distributes further by diffusion [10,16].

Mechanical factors important to CED are viscosity of infusate, infusion flow rate, and volume of infusion. In general, the volume of distribution is proportional to the volume of infusion [10,17,18]. Intrinsic molecular properties including size, polarity, binding to extracellular matrix proteins and or enzymes and export through efflux pumps all contribute to the convection volume and persistence of therapy [10,17,18]. Effective flow rates range between 1 and 10 µL/min as higher flow rates are susceptible to backflow and poor convection.

### 3.2. CED Logistics

Prior to CED, advanced neuroimaging including computed tomography (CT) and magnetic resonance imaging (MRI) are necessary. MR imaging should at minimum include T1, T2, and T1 post-contrast images, which is especially true if gadolinium or another contrast agent is to be injected with the infusate. Additional methods including susceptibility weighted imaging, diffusion weighted imaging, or T2-gradient echo and BOLD-fMRI are helpful in better characterization of necrosis, edema, and functional neuroanatomic structures. In addition, all can help with catheter trajectory planning [16,19]. Tumor volume, areas of necrosis, potential flow voids, proximity to ventricles, peritumoral edema, and vasculature or breaches into the ventricular system are all critical variables for determining the catheter tip placement and optimal convection volume [16].

Historically, the CED infusion time has spanned a few hours to at most a few days. Most studies have had infusions lasting at maximum 96 h. These infusions have used an externalized catheter connected to an infusion pump. Patients generally remain as inpatients in order to help minimize the risk for infection and to facilitate catheter removal. Recent studies have used catheters attached to transcutaneous ports to allow for the infusion which occurs to outpatients, as no externalized hardware is present [20]. Our group has also attempted to address the chronicity but through the use of a subcutaneous implantable pump, as described below [21].

### 3.3. Pitfalls of CED

Two key considerations that impact the efficacy of CED are backflow or reflux and the presence of pathology near the catheter tip. Reflux or backflow of the infusate can alter this relationship and dramatically diminish the volume of distribution [10,16,22]. Backflow occurs when a pocket forms around the catheter due to mechanical shearing during catheter placement or pressure spikes due to the flow rate. Mechanical shearing can be minimized through the use of soft and thin catheters [23]. Stepped profile cannulas have been shown to limit the reflux even with high flow rate infusions. This design combines a wide bore cannula with a narrow tip attached by a sharp transition or step [24,25]. The presence of pressure spikes can be minimized by the use of porous catheters with multiple infusate outlets preventing occlusions [23].

While systemic toxicity is greatly reduced, CED can be associated with certain complications. A risk of infection (i.e., bacterial meningitis, subdural empyema, abscess at the catheter tip) is rare but has been reported [26]. More common dose limiting toxicities include chemical meningitis or seizure highlighting the need for studies identifying the maximum tolerated dose [27].

Factors related to brain pathology also impact distribution patterns with CED. Areas of necrosis with a lack of interstitial architecture can lead to infusate pooling. Highly vascular regions—common in high-grade glioma—can lead to infusate leaking into systemic circulation [28]. In a phase I trial of CED of the Delta-24-RGD adenovirus in recurrent GBM patients, the treatment infusion was preceded by gadolinium infusion. These studies demonstrated that the leakage of gadolinium into the CSF was associated with the decreased volume of distribution to volume of infusion ratios. Furthermore, they also showed that intratumoral catheter placement led to the decreased volume of distributions likely due to tumor vasculature and necrosis [29]. These considerations are important when combining catheter placement and surgical resection for high grade glioma patients.

## 4. Convection Enhanced Delivery—The Columbia University Medical Center Experience

### 4.1. Preclinical Data—Small Animal Models

At our institution, we have repeatedly demonstrated the efficacy of CED in treating orthotopic models of glioblastoma. Our investigational therapies have focused on topoisomerase inhibitors, specifically topotecan (TPT). TPT has a long clinical history demonstrating safety in local and systemic delivery [21,30,31,32,33]. The presence of dose limiting systemic toxicities make it an optimal candidate for local delivery. Moreover, as a topoisomerase 1 inhibitor, it is toxic to glioma cells and relatively non-toxic to the normal brain tissue [34].

The first TPT CED study involved treating rats with orthotopically implanted C6 glioma cells. These mice were treated with intracerebral infusion of topotecan (TPT) through a CED microcatheter. Both a high dose (160 µg/kg/day) and low dose (32 µg/kg/day) CED of topotecan led to complete tumor regression and cure. Notably one mouse in the high-dose group did die from neurological toxicity [34]. This preliminary study showed that TPT CED was safe as animals tolerated the infusion without decreases in weight. More importantly it showed that TPT CED was effective as 11/12 mice became long-term survivors [34]. These results were replicated in an orthotopic syngeneic rat model that relies on retroviral infection with a PDGF-IRES-GFP construct. Infected cells overexpress the platelet derived growth factor (PDGF) and reliably form tumors with characteristic features of glioblastoma including endothelial proliferation, pseudopalisading necrosis, and hypercellularity [35]. Topotecan at a dose of 136 uM was infused for 1, 4, or 7 days. Compared to the control, all topotecan infusion groups had improved survival. Importantly, rats treated with 7 days of topotecan CED had a 1.7× increase in the median survival over rats treated with 4 days of topotecan CED (Figure 1) [35]. The correlation between the prolonged treatment duration and treatment efficacy likely relates to the TPTs mechanism of action. Topotecan is a topoisomerase I inhibitor and thus acts during the S-phase of the cell cycle. The greater duration of treatment ensures that more cells will enter the S-phase and experience the cytotoxic effects of TPT [36].

We have also tested another topoisomerase inhibitor, etoposide, in an orthotopic syngeneic murine model of glioma. This model, which is also driven by PDGF overexpression, shows close transcriptional similarity with human proneural glioma [37,38]. CED with a high dose (80 µM) etoposide for 7 days led to a significant survival benefit with half of the animals becoming long-term survivors; a low dose treatment (4 µM) led to no survival benefit [37]. These findings point to two key factors that drive tumor response: High local concentrations and prolonged duration of treatment.

### 4.2. Pre-Clinical Data—Large Animal Models

Based on our murine and rodent studies, we hypothesized that prolonged or chronic TPT CED would enhance the therapeutic effect in glioma patients. To this end, we developed a mechanism for an implantable subcutaneous pump (Synchromed II, Medtronic; Minneapolis, MN). We first used a large animal pig model to demonstrate that the subcutaneous pump was safe, and allowed drug distribution for periods ranging from 3–10 days (Figure 2) [21]. We were further able to broaden this pig model by expanding the limits of chronic convection enhanced delivery using an implanted pump. We utilized a total of 12 pigs in which we performed CED using an implantable subcutaneous pump with varying convection durations ranging from 4 to 32 days. In both experiments, TPT concentrations of 136 µM were used. Neurobehavioral side effects were measured showing minimal measurable side effects of prolonged treatment.

### 4.3. Gadolinium as a Surrogate for Convection Volumes

The noninvasive methodology for monitoring drug distribution is critical for a successful application of CED in the clinic. To model this in our pig experiments gadodiamide was added to the TPT infusate. For an accurate evaluation of the convection treatment volume, the subcutaneous pump was loaded with a mixture of TPT and gadodiamide. Gadodiamide has a molecular weight of 591.7 g/mol and is freely water soluble, while TPT has a molecular weight of 421.4 g/mol and a solubility in water of 1 mg/mL. Topotecan is a quinolone alkaloid derivative with numerous hexacyclic rings [39,40]. Even with these different physical properties, our data suggest that TPT and gadodiamide distribute in similar patterns when co-infused using CED. We showed that co-infusion of a contrast agent with topotecan would allow gadolinium signal intensity to act as an accurate surrogate for topotecan concentration (Figure 3) [41]. This model also helped determine the time course associated with convection volumes. During the long-term infusions, a maximum contrast enhancement was reached by day 3 or 4. The greatest increase in volume of distribution occurred during the first 48 h. Over the 32 day infusion course, the volume of distribution would decline from the maximum volume until a steady state was achieved. Local anticipated signal hypodensities at the catheter tip were also observed with both changes likely attributable to the local tissue and tumor response to infusion. Again, such radiographic changes were not associated with the neurological deficit or evidence of toxicity [41].

### 4.4. Clinical Experience with Topotecan

Based on the success of TPT by CED in our animal models, we received FDA approval to conduct a clinical trial in patients with recurrent malignant gliomas. Our initial Phase IB clinical trial studied 10 glioblastoma and six anaplastic astrocytoma patients in a dose escalation study to determine the maximum tolerated dose. Infusions of topotecan continued for 100 h with a flow rate of 200 µL/h. Four dose levels were studied: 0.04, 0.0667, 0.1, and 0.133 mg/mL. We did observe two events of dose limiting toxicity that established the maximum tolerated dose. Eleven out of 16 patients demonstrated either early response or pseudoprogression showing that the infusion of topotecan could lead to tumor specific cell death with minimal adverse events [42]. Patients who had an early response or pseudoprogression on MRI had a significantly improved overall survival. GBM patients in this cohort had a 20%, 2-year survival following the treatment. Two patients became long-term survivors with a survival from treatment of over 10 years [42]. To date, one patient from the cohort remains alive (Figure 4). This finding coupled with the high percentage of patients with tumor response demonstrated that further optimization of treatment regimens may require chronic dosing of therapies through the CED catheter.

This patient cohort was evaluated using the HeadMinder Cognitive Stability Index (CSI) and the SF-36 Health Survey (SF-36) at baseline and 4, 8, 12, and 16 weeks post-treatment. These scales tested a host of neurocognitive functions including processing speed, spatial memory, and working memory. The quality of life assessments focused on physical pain, mental health, emotional health, and general vitality. Across both investigatory questionnaires, most patients reported stable or improved scores functioning across these areas at 4 weeks; over 75% of patients reported stable or improved scores at 8 and 16 weeks post-treatment [43].

We also explored the use of TPT CED in the treatment of DIPG. Our experience with two pediatric patients showed that the catheter placement and TPT infusion is technically feasible. We also observed that high infusion rates and high infusion volumes were associated with functional decline. This finding highlights the need for chronic treatment schedules for minimization of potential harms [44].

### 4.5. Current Chronic CED Clinical Studies

Based on our preclinical data showing that the efficacy of TPT with CED in gliomas improves with the increased treatment duration, we designed a clinical protocol that utilizes the implanted pump strategy validated in our pig model. We received FDA approval to conduct a Phase IB clinical trial in five human GBM patients, using chronic convection enhanced delivery of topotecan through an implantable and programmable subcutaneous pump (Synchromed II, Medtronic; Minneapolis). There were multiple goals of this trial. Primarily, we wanted to prove that chronic convection enhanced delivery via an implantable pump is a safe technique for drug delivery in humans. Furthermore, we wanted to validate the use of gadolinium as a surrogate marker for topotecan drug concentration when co-infused through the convection enhanced delivery catheter. Finally, we wanted to utilize our expertise with MRI-localized biopsies to get both histologic and molecular based analyses of pre-treatment and post-treatment samples to understand the various effects drug infusion can have on the tumor and tumor microenvironment. The full schematic of the clinical trial can be found in Figure 5. The trial was recently concluded and the results are currently being analyzed.

### 4.6. Best Practices

Our experience has led us to develop a set of best practices for CED clinical trials. Initially, compounds of interest need to be tested in a large animal model prior to initiation of the clinical trial. The most important reason for this is to validate the drug distribution parameters and their correlation with radiological findings such as gadolinium enhancement while proving safety. Newer techniques such as MR spectroscopy or PET imaging may also be important to validate for different classes of therapeutic molecules [45,46,47]. Neurobehavioral testing including performance outcomes are critical for determining dose limiting toxicities associated with CED. Finally, evaluation of the chronicity of treatment should be undertaken in relevant preclinical models to help guide the creation of optimal trial guidelines.

## 5. Current Landscape of CED Clinical Treatments

There has been an increase in the number of clinical trials utilizing CED. The first study examining CED in malignant brain tumors utilized a transferrin conjugated to a genetically mutated diphtheria toxin [48]. Since then, multiple studies have used chemotherapies such as carboplatin, paclitaxel, or topotecan in patients with recurrent glioma [26,42,49]. These chemotherapies were often found to be too toxic when delivered systemically and are having a renaissance with local delivery. Since 2013, multiple groups including ours have utilized CED for the delivery of chemotherapy in adult brainstem gliomas and pediatric diffuse infiltrating gliomas [44,50,51,52]. Beyond chemotherapies, CED is being used for targeted delivery of cytotoxins with toxin conjugates. These cytotoxins generally work by targeting a surface receptor overexpressed in glioma cells with a protein which is then conjugated to an endotoxin. Phase I and II trials have targeted the IL13 receptor, IL4 receptor, TGF-alpha receptor, and the CD155 receptor to name a few [53,54,55,56]. Desjardins et al. reported on a Phase I clinical trial of 61 patients treated with a dose escalation of a recombinant polio-rhinovirus chimera, which has a particular tropism for CD155 [57]. The other studies have conjugated a cytokine or protein of interest (IL13, IL4, TGF-α) to the pseudomonas exotoxin, thereby targeting this toxin to glioma cells leading to glioma cell specific toxicity [55,58]. Phase II studies have focused on immunological modifiers examining local CED of CpG oligonucleotides as immune agonists [59,60]. Antibody therapies are also being delivered via CED; clinical studies have used radiolabeled antibodies against CD276 in DIPG patients [50], while pre-clinical studies span the gamut from validated antibodies including, cetuximab, bevacizumab to novel designer constructs [61,62,63,64]. A summary of clinical trials using CED for glioma—based on a NCBI pubmed search for “convection enhanced delivery glioma” and “clinical trial”—and their clinical trial parameters can be found in Table 1.

These studies highlight two key benefits of CED: The ability to avoid systemic toxicity and maximize locoregional dosing of therapy. Additionally, CED allows the use of new classes of drugs including proteins and novel biologics that would not be feasible with systemic delivery. In all the studies described, however, a true survival benefit remains elusive even with documented tumor regression and treatment response. Even though the benefits of CED have been thoroughly described, one key limitation was the duration of the treatment. Our success prolonging the treatment duration with implantable pumps requires an additional study to overcome current treatment limitations. Of all the published trials, only the trial outlined by Bogdahn et al. utilizes chronic treatment parameters. This trial connected a CED catheter to a subcutaneous port access system and utilized an external pump to allow patients to receive up to 11 treatment cycles in an outpatient setting [65].

## 6. Investigational Drug Formulation

As described, CED provides a mechanism to improve drug delivery across the BBB. Drug features such as hydrophilicity or size can still impact the distribution through the interstitial space. To optimize drug delivery therapeutics have been combined with various nanoparticle formulations including but not limited to: Liposomes, micelles, and polymeric nanoparticles [71]. The goal of nano-encapsulation of drugs for delivery is to protect compounds from enzymatic degradation or efflux, decrease toxicity, and improve drug targeting and distribution [71]. Many pre-clinical studies have used poly-ethylene glycol (PEG) coated nanoparticles to improve the brain penetration of therapies such as paclitaxel [72,73]. Importantly, PEG can decrease the tumor cell uptake of therapies. The impact of nano-encapsulation on the uptake of therapies needs to be evaluated on a case-by-case basis for each type of nano-encapsulation.

Beyond their applicability in drug delivery, nanoparticles can be used to deliver contrast or radioactive material for accurate MRI or CT imaging of convection volume. These methods have been demonstrated in small and large animal models and multiple clinical trials are ongoing to investigate nanoparticle loading with CED [74,75]. Nanoparticle formulations of panobinostat, a notoriously hydrophobic and unstable drug, are being investigated in CED clinical trials for DIPG patients [76,77].

## 7. Potential Applications of CED beyond Glioma Therapy

The consistent safety profile seen in CED clinical trials of glioma patients has laid the groundwork for CED to be used in disease processes where systemically delivered drugs have been ineffective. The ability for targeted chronic delivery of compounds—ranging from biologics to small molecules to viruses—directly into the brain parenchyma can have a profound impact on a host of other diseases. Numerous studies have demonstrated the efficacy of CED in animal models of Alzheimer’s disease. These studies in mice and rats use a host of different compounds from bacteriophages, to amyloid degrading enzymes, to viral vectors and antibodies [78,79,80,81]. Moreover, more recent studies show that the efficacy with systemic delivery of antioxidant compounds may benefit from higher concentrations, which could be achieved with local delivery [82,83]. Researchers have also validated the use of CED in non-human primates showing that consistent targeting to deep brain structures, such as the entorhinal cortex or hippocampus, is feasible and safe [84].

The use of CED in Parkinson’s disease (PD) is another burgeoning field. Pre-clinical studies have demonstrated efficacy using CED for the delivery of adenoviral based gene therapies and for the delivery of compounds that prevent oxidative stress and subsequent neurodegeneration [85,86,87]. Phase I experiments delivering glial derived neurotrophic factors or an adenoviral vector delivering aromatic L-amino-acid decarboxylase for PD patients have been undertaken. Although safety has been established with CED to deep brain structures, larger studies are necessary to describe the outcome improvements [88,89,90]. These studies all highlight the potential benefit that CED can have in many disease processes outside of the glioma treatment.

## 8. Conclusions

CED is a safe, effective, and feasible method for direct intraparenchymal delivery of anti-tumor compounds. It provides a novel and measurable strategy to treat patients with recurrent glioblastoma by achieving a local-regional distribution of a drug directly into the tumor and the surrounding brain bypassing the inherent limitations imposed by conventional systemic delivery. Co-infusion of Gd provides a reliable, measurable, and non-invasive method for monitoring drug distribution in real time. The optimization of this delivery technique in glioma patients now provides the opportunity for these same techniques to be applied to other pathological states. The field has continued to develop with catheters, which are able to minimize backflow, maximize volume of distribution, and even allow for chronic treatment over days to weeks. Future studies may seek to determine safe methods for refillable CED pumps, which can allow for multiple drug treatments spread out in time through a single catheter.

## Figures and Tables

**Figure 1 pharmaceutics-13-00039-f001:**
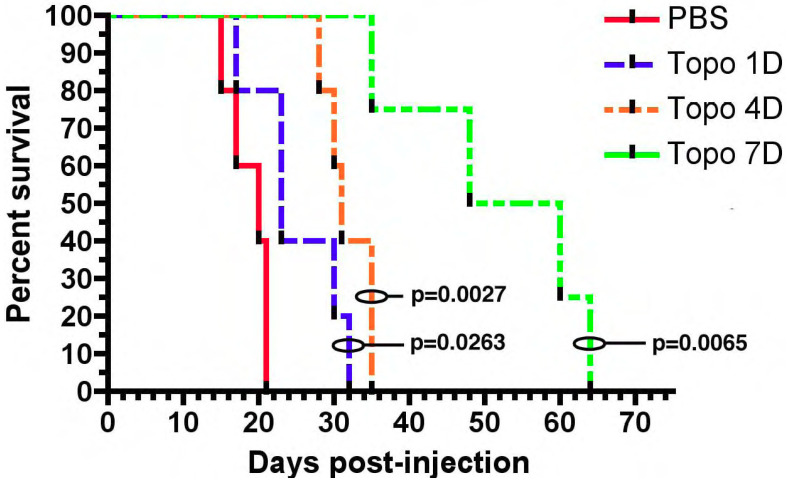
Chronic convection enhanced delivery (CED) of topotecan provides a significant survival advantage. Rats with orthotopic virally-induced, platelet derived growth factor (PDGF) driven tumors were treated with CED of topotecan for various days (1, 4, or 7 days). *p*-values show a comparison to the PBS control. 1, 4, and 7 days CED survival were each significantly greater than PBS (*p* < 0.05). The median survival for 7 days CED was significantly greater than the other groups, (median survival: PBS—20 dpi, 1 day—23 dpi, 4 day—31 dpi, 7 day—54 dpi; *p* < 0.05). The figure is reproduced with permission from [35], Cancer Res. 2011.

**Figure 2 pharmaceutics-13-00039-f002:**
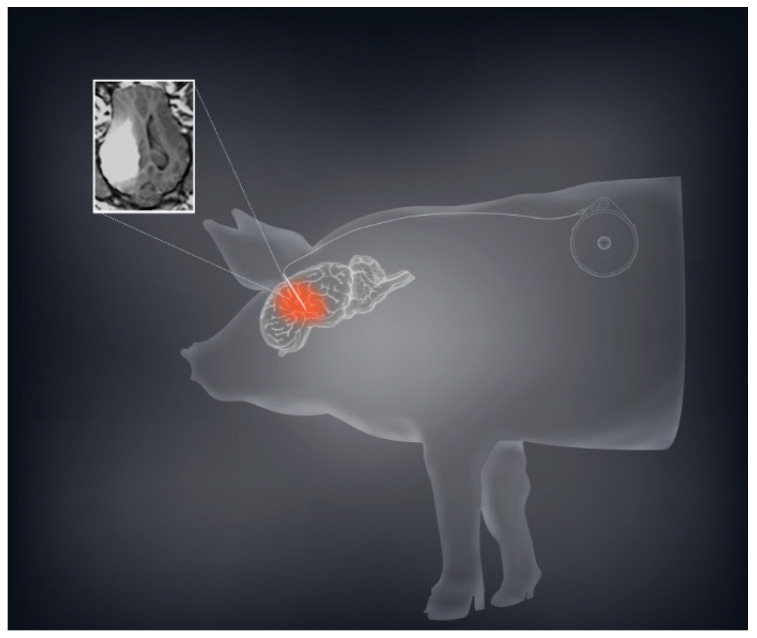
CED using a subcutaneous implantable pump is safe and feasible in a large animal model. A schematic demonstration with the sample MRI of convection volume with co-infusion of topotecan and gadolinium is shown demonstrating a sizeable volume of distribution.

**Figure 3 pharmaceutics-13-00039-f003:**
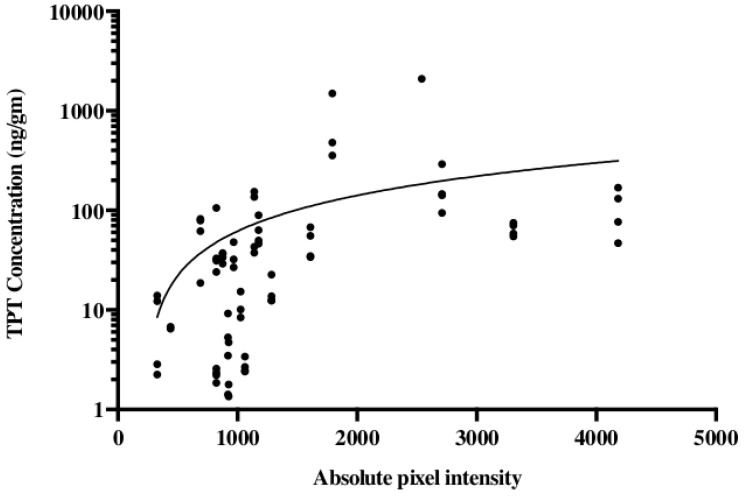
Topotecan concentration is correlated with the gadolinium absolute pixel intensity of T1 MRI. Studies were conducted in a porcine model using a subcutaneous pump that co-delivered topotecan and gadolinium. Liquid chromatography-mass spectrometry was used for the quantification of topotecan concentration and correlated with the absolute pixel intensity from T1-weighted MRI. The data are reproduced from D’Amico et al., 2019 [41].

**Figure 4 pharmaceutics-13-00039-f004:**
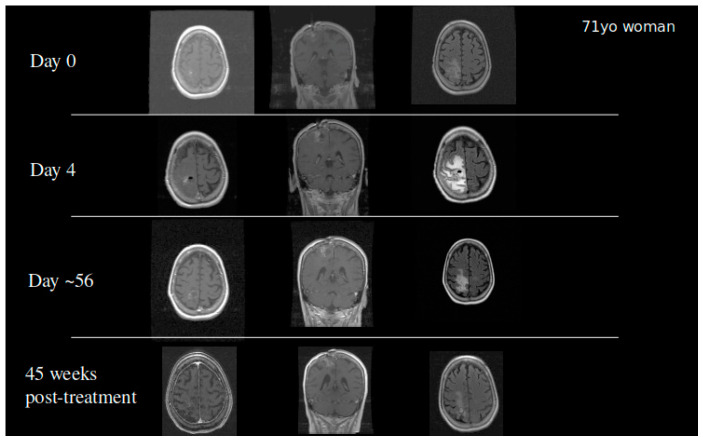
A 71-year-old woman who developed a parietal lobe syndrome on the fourth day of topotecan (TPT) by the CED treatment. Scans are shown at the time of treatment (day 0), at the time the dose limiting toxicity occurred (day 4), 8 weeks after treatment, and 45 weeks after treatment with approximately 90% return to her baseline neurological exam. She was progression free 3 years after the treatment and died just under 4 years post CED.

**Figure 5 pharmaceutics-13-00039-f005:**
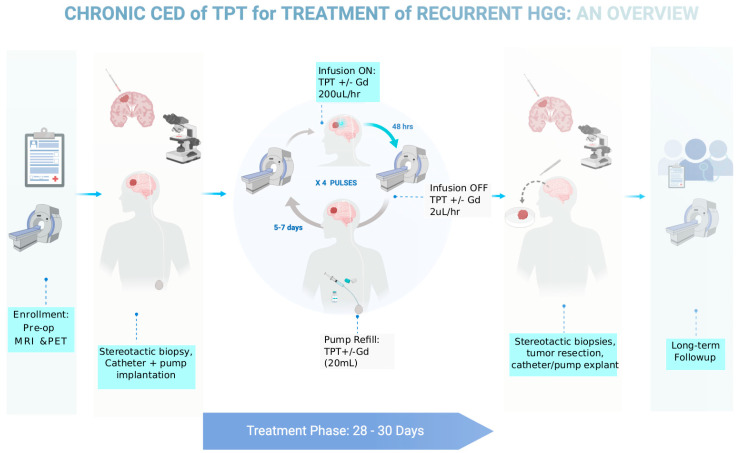
Diagram outlining the clinical trial parameters for a recently completed Phase IB trial for chronic CED of topotecan in patients with recurrent glioblastoma.

**Table 1 pharmaceutics-13-00039-t001:** Summary of clinical studies.

Author	Description	Investigational Drug	Device(s) Used	Flow Rate/Time	Results
Heiss et al. 2018 [51]	Phase I study of five pediatric DIPG patients. NCT00088061	IL13-Pseudomonas toxin	Navigus catheter guide, Medfusion 3500 pump	5–10 uL/min, maximum 13 h	Infusions led to transient CN deficits. All patients had disease progression by 12 weeks post-infusion.
Desjardins et al. 2018 [57]	Phase I dose escalation in 61 patients with recurrent GBM. NCT01491893	PVSRIPO	Vygon PIC030 Sophysa catheter, Medfusion 3500 pump	500 ul/h, 6.5 h	21% of patients survived at 24 and 36 months.
Souweidane et al. 2018 [50]	Phase I dose escalation study in 28 pediatric DIPG patients. NCT01502917	^124^I-8H9	SmartFlow cannula	Max 7.5 uL/min	No dose limiting toxicities observed. Across the seven doses, a significant dose dependent increase in overall survival was observed.
Anderson et al. 2012 [44]	Phase IB in two pediatric patients with DIPG. NCT00308165	Topotecan	Silastic infusion catheter, Medfusion 2010 pump	<0.04 mL/h, V_i_ 5–6 mL over 100 h	Brainstem CED is feasible. Flow rates over 12 mL/h were associated with new neurological deficits.
White et al. 2012 [49]	Phase I dose escalation study in 18 patients with recurrent or progressive GBM	Carboplatin	In-house catheter	8 or 16 h	N/A
Bogdahn et al. 2011 [65]	Phase IIB for 145 patients with recurrent GBM/AA	Trabedersen (TGF-B2 inhibitor)	Subcutaneous port access system with external pump	1 uL/min up, up to 11 treatment cycles	Trend towards a survival benefit for CED of 10 uM Trabedersen (39.1 months), 80 uM Trabedersen (35.2 months), chemotherapy alone (21.7 months).
Bruce et al. 2011 [40]	Phase IB dose escalation study of 10 GBM patients and six AA patients. NCT00308165	Topotecan	Silastic infusion catheter, 2.5 mm diameter	200 uL/h for 100 h	Six month PFS for GBM patients was 44%, AA patients 75%. Sixty-nine percent demonstrated radiographic response.
Kunwar et al. 2010 [66]	Phase III trial of 296 recurrent GBM patients. NCT00076986	CB (IL13-PE) vs. gliadel wafers	2–4 catheters used	0.75 mL/h over 96 h	Median survival was 45.3 weeks for CED patients and 39.8 weeks for gliadel patients (*p* = 0.310).
Carpentier et al. 2010 [61]	Phase II trial in 34 recurrent GBM patients. NCT00190424	CpG: Immunostimulating motifs	Seldiflex and Plastimed catheters	3.3 uM/h over 6 h	6 month PFS: 19%. Median OS 28 weeks.
Sampson et al. 2008 [67]	Phase I dose escalation study of 20 recurrent glioma patients	TGF-alpha PE (TP38)	PS Medical CSF ventricular catheter, outer diameter 2.1 mm	0.4 mL/h over 50 h	Co-infusion with ^123^I-albumin showed V_d_ of > 4 cm away from catheter. A total of 2/15 patients with radiographic response.
Vogelbaum et al. 2007 [68]	Phase I trial in 21 patients with newly diagnosed GBM. NCT00089427	CB (IL13-PE)	Vygon Neuro: 1 mm inner diameter barium impregnated catheter	0.75 mL/h over 96 h	No dose limiting toxicities at a dose of 0.25 ug/mL with EBRT and TMZ.
Kunwar et al. 2007 [69]	Phase I trial in 51 GBM and AA patients.	CB (IL13-PE)	N/A	0.75 mL/h over 96 h	Patients with optimally placed catheters survived longer (55.6 vs. 37.4 weeks *p* = 0.03).
Sampson et al. 2007 [70]	Phase I study of seven patients with recurrent malignant glioma	CB + 123I-labeled albumin	Vygon Neuro - 1 mm inner diameter barium impregnated catheter	0.540–0.750 mL/h over 96 h	SPECT demonstrated that intratumoral infusion led to poor volume of distribution.
Carpentier et al. 2006 [59]	Phase I study with 24 recurrent GBM patients	CpG immuostimulatory oligonucleotides	N/A		Median survival 7.2 months with only one dose limiting toxicity observed.
Popperl et al. 2005 [45]	Phase I study with eight recurrent GBM patients	Paclitaxel	Silicone catheter, Medfusion 2010 pump	0.3 mL/h over 120 h	FET-PET uptake correlates with tumor burden and decreased following CED.
Lidar et al. 2004 [26]	Phase I/II trial of 15 recurrent GBM/AA patients	Paclitaxel	VPS Medtronic silicone catheter, Medfusion 2010 pump	0.3 mL/h over 120 h	DW-MRI showed five complete responses and six partial responses out of 15 cases. Responses were confirmed via biopsies and en-bloc resection.
Sampson et al. 2003 [56]	Phase I dose escalation study of 20 recurrent malignant glioma patients	TGF-alpha PE (TP38)	N/A	0. 4 mL/h over 50 h	A total of 3/15 patients with radiographic response and 4/20 have no evidence of tumor recurrence.
Weber et al. 2003 [54]	Phase I dose escalation of 25 GBM and six AA patients NCT00003842	IL-4 PE	N/A	96 h infusion	Median survival of GBM patients was 5.8 months following treatment. Thirty-nine percent of patients had drug related toxicity.
Laske et al. 1997 [48]	Phase I dose escalation of 10 GBM, five AA, one AO, and one LC patients	Tf-CRM107	Silastic infusion catheters (2.5 mm outer diameter), Medfusion 2001 syringe pump	4–10 µL/min over 4 h	A total of 9/15 patients had 50% reduction in tumor volume with two complete responses.

GBM: Glioblastoma; AA: Anaplastic astrocytoma; AO: Anaplastic oligodendroglioma; LC: Lung adenocarcinoma; CB: Cintredekin besudotox; IL13-PE: IL-13 pseudomonas exotoxin; CN: Cranial nerve; PVSRIPO: Polio-rhinovirus chimera; TMZ: Temozolomide; EBRT: External beam radiation therapy; SPECT: Single photon emission computed tomography; TGF: Transforming growth factor; V_d_: Volume of distribution; Tf-CRM107: Mutated Transferrin linked to diphtheria toxin.

## Data Availability

Data for this review is publicly available through previously published manuscripts as referenced.

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
