# Peer review of "Convection Enhanced Delivery of Topotecan for Gliomas: A Single-Center Experience"

_pharmaceutics, 2020, doi:10.3390/pharmaceutics13010039_

Round 1
Reviewer 1 Report
Upadhyayula and colleagues proposed a very interesting review article on the application of Convection Enhanced Delivery (CED) for the treatment of glioma. The authors well described the mechanisms behind CED and all the relevant information on CED and TPT administration. In addition, the authors provided also interesting data about the current and potential application of CED. Overall, the manuscript is well written and the structure of the manuscript is appropriate. Below are reported some minor comments that will improve the quality of the manuscript:
1) Please provide references supporting the following statements: “Current standard of care consisting of resection, temozolomide3 and radiation 4, epitomizes the scarcity of major breakthroughs in the past 50 years and the desperate need for more comprehensive and innovative approaches to make transformative contributions in the clinical management of GBM. Although numerous chemotherapeutic drugs demonstrate significant anti-tumor activity in preclinical studies, their efficacy in clinical trials has been disappointing because systemic delivery fails to achieve therapeutic drug levels in tumor cells due blood-brain barrier permeability and systemic toxicity.”. For this purpose, see:
- 10.3389/fphar.2018.01300
- 10.1007/s11060-008-9774-3
2) In Chapter 3 “Convection Enhanced Delivery - Rationale and Mechanisms”, the authors should indicate how long a CED treatment lasts, if it needs hospitalization, or preparative procedures. Please, add this important information;
3) In the subheading “Pitfalls of CED” of Chapter 4, please indicate if bacterial infection at the cannula entry point can occur;
4) In Figure 1, it is not clear if the p-values are related to PBS vs Topo1D, PBS vs Topo4D and PBS vs Topo7D. In addition, it is important to determine the statistical differences existing between Topo1D vs Topo4D, Topo1D vs Topo7D and so on. Please provide these data;
5) In the subheading “Gadolinium as a Surrogate for Convection Volumes” the authors should clarify if gadolinium and TPT have similar chemical properties. Indeed, the distribution of the two molecules depends on their molecular weight, hydrophilicity, etc. Please, argue this point;
6) In chapter 5, please indicate if CED study using monoclonal antibodies have been performed;
7) If there are available data, chapter 6 should be extended describing the use of drugs for Parkinson’s disease, schizophrenia and other neurological disorders.
Author Response
Upadhyayula and colleagues proposed a very interesting review article on the application of Convection Enhanced Delivery (CED) for the treatment of glioma. The authors well described the mechanisms behind CED and all the relevant information on CED and TPT administration. In addition, the authors provided also interesting data about the current and potential application of CED. Overall, the manuscript is well written and the structure of the manuscript is appropriate. Below are reported some minor comments that will improve the quality of the manuscript:
We thank the reviewer for their time and in-depth review.
1) Please provide references supporting the following statements: “Current standard of care consisting of resection, temozolomide3 and radiation 4, epitomizes the scarcity of major breakthroughs in the past 50 years and the desperate need for more comprehensive and innovative approaches to make transformative contributions in the clinical management of GBM. Although numerous chemotherapeutic drugs demonstrate significant anti-tumor activity in preclinical studies, their efficacy in clinical trials has been disappointing because systemic delivery fails to achieve therapeutic drug levels in tumor cells due blood-brain barrier permeability and systemic toxicity.”. For this purpose, see:
- 10.3389/fphar.2018.01300
- 10.1007/s11060-008-9774-3
We agree that citations for this statement were overlooked. We have added the citations recommended.
2) In Chapter 3 “Convection Enhanced Delivery - Rationale and Mechanisms”, the authors should indicate how long a CED treatment lasts, if it needs hospitalization, or preparative procedures. Please, add this important information;
We have added this information in a section titled “CED Logistics”. The section reads:
“Prior to CED, advanced neuroimaging including computed tomography (CT) and MRI are necessary. MR imaging should at minimum include T1, T2 and T1 post-contrast images; this is especially true if gadolinium or other contrast agent is to be injected with the infusate. Additional methods including susceptibility weighted imaging, diffusion weighted imaging or T2-gradient echo and BOLD-fMRI are helpful in better characterization of necrosis, edema, and functional neuroanatomic structures and all can help with catheter trajectory planning.16,19 Tumor volume, areas of necrosis, potential flow voids, proximity to ventricles, peritumoral edema and vasculature are all or breaches into the ventricular system are all critical variables for determining catheter tip placement and optimal convection volume.16
Historically, CED infusion time has spanned a few hours to at most a few days. Most studies have had infusions lasting at maximum 96 hours. These infusions have used an externalized catheter connected to an infusion pump. Patients generally remain inpatient to help minimize the risk for infection and to facilitate catheter removal. Recent studies have used catheters attached to transcutaneous ports to allow for infusion to occur outpatient as no externalized hardware is present.20 Our group has also attempted to address the chronicity but through use of a subcutaneous implantable pump, as described below.21”
3) In the subheading “Pitfalls of CED” of Chapter 4, please indicate if bacterial infection at the cannula entry point can occur;
This is an important point. More broadly we did not mention general complications in CED. We have added the following:
“While systemic toxicity is greatly reduced, CED can be associated with certain complications. A risk of infection (ie. bacterial meningitis, subdural empyema, abscess at catheter tip) is rare but has been reported.26 More common dose limiting toxicities include chemical meningitis or seizure highlighting the need for studies identifying the maximum tolerated dose.27”
4) In Figure 1, it is not clear if the p-values are related to PBS vs Topo1D, PBS vs Topo4D and PBS vs Topo7D. In addition, it is important to determine the statistical differences existing between Topo1D vs Topo4D, Topo1D vs Topo7D and so on. Please provide these data;
This data has been added to the Figure 1 legend. The legend now reads:
“Figure 1. Chronic CED of Topotecan provides a significant survival advantage. Rats with orthotopic virally-induced, PDGF driven tumors were treated with CED of Topotecan for various days (1 day, 4 day or 7 days). P-values show comparison to PBS control. 1D, 4D and 7D CED survival were each significantly greater than PBS (p<0.05). Median survival for 7 days CED was significantly greater than other groups, (Median Survival: PBS - 20 dpi, 1 day - 23 dpi, 4 day - 31 dpi, 7 day - 54 dpi; p<0.05).”
5) In the subheading “Gadolinium as a Surrogate for Convection Volumes” the authors should clarify if gadolinium and TPT have similar chemical properties. Indeed, the distribution of the two molecules depends on their molecular weight, hydrophilicity, etc. Please, argue this point;
This point is well taken. Although direct comparisons of these properties are tought to find we have described some chemical properties of each molecule with the following sentences:
“For accurate evaluation of the convection treatment volume, the subcutaneous pump was loaded with a mixture of TPT and gadodiamide. Gadodiamide has a molecular weight of 591.7g/mol and is freely water soluble while TPT has a molecular weight of 421.4g/mol and a solubility in water of 1mg/mL. Topotecan is a quinolone alkaloid derivative with numerous hexacyclic rings.39,40”
6) In chapter 5, please indicate if CED study using monoclonal antibodies have been performed;
To address this we have added the following sentence:
“Antibody therapies are also being delivered via CED; clinical studies have used radiolabeled antibodies against CD276 in DIPG patients,48 while pre-clinical studies span the gamut from validated antibodies including, cetuximab, bevacizumab to novel designer constructs.59–62”
7) If there are available data, chapter 6 should be extended describing the use of drugs for Parkinson’s disease, schizophrenia and other neurological disorders.
We were able to find multiple papers pre-clinical and clinical studying parkinson’s disease and have added the following section to address this comment. Unfortunately data for schizophrenia or other neurological/neuropsychiatric disorders was substantially more limited and more theoretical.
“Use of CED in Parkinson’s disease (PD) is another burgeoning field. Pre-clinical studies have demonstrated efficacy using CED for delivery of adenoviral based gene therapies and for delivery of compounds that prevent oxidative stress and subsequent neurodegeneration.78–80 Phase I experiments delivering glial derived neurotrophic factor or an adenoviral vectors delivering aromatic L-amino-acid decarboxylase for PD patients have been undertaken. Although safety has been established with CED to deep brain structures, larger studies are necessary to describe outcome improvements.81–83 These studies all highlight the potential benefit that CED can have in many disease processes outside of glioma treatment.”
Reviewer 2 Report
The authors reviewed a convection-enhanced delivery for glioma. This review manuscript is interesting, however, there are several issues to be revised.
- Several clinical studies were summarized in table 1. Also, some clinical trials were conducted in the authors’ institution. The NCT number should be shown if these trials had the number.
- The size of characters was small in the figure 5. The authors should make the figure clearly.
Author Response
We thank the reviewer for their time and comments.
- Several clinical studies were summarized in table 1. Also, some clinical trials were conducted in the authors’ institution. The NCT number should be shown if these trials had the number.
We have addressed this concern. Notably of the trials included, roughly half don't have an identifiable NCT number. In these cases no NCT was listed in the manuscript and after searching through clinicaltrials.gov it was not possible to ascertain exactly which trial corresponded to which manuscript. This usually occurred because multiple trials with similar agents and descriptions were listed encompassing many of the same institutions from each manuscript. In many cases we used similar word choice or follow-up parameters to determine which clinical trial corresponds to which manuscript.
- The size of characters was small in the figure 5. The authors should make the figure clearly.
We have remade figure 5 with larger text for increased readability.
We once again thank the reviewer for their time.
Reviewer 3 Report
The review is well-organized. We suggest the authors can input some new formulation for CED. For example, liposome or polymeric micelles.
On the other hand, it is desired to describe some future application or perspective viewpoint.
Author Response
We thank the reviewer for their time and comments.
The review is well-organized. We suggest the authors can input some new formulation for CED. For example, liposome or polymeric micelles. On the other hand, it is desired to describe some future application or perspective viewpoint.
This point is well taken. We have added a section titled "Investigational Drug Formulation" to address these comments. The section reads:
As described, CED provides a mechanism to improve drug delivery across the BBB. Drug features such as hydrophilicity or size can still impact distribution through the interstitial space. To optimize drug delivery therapeutics have been combined with various nanoparticle formulations including but not limited to: liposomes, micelles and polymeric nanoparticles.55 The goal of nano-encapsulation of drugs for delivery is to protect compounds from enzymatic degradation or efflux, decrease toxicity, and improve drug targeting and distribution.55 Many pre-clinical studies have used poly-ethylene glycol (PEG) coated nanoparticles to improve brain penetration of therapies such as paclitaxel.56,57 Importantly PEG can decrease tumor cell uptake of therapies. The impact of nano-encapsulation on uptake of therapies needs to be evaluated on a case-by-case basis for each type of nano-encapsulation.
Beyond their applicability in drug delivery, nanoparticles can be used to deliver contrast or radioactive material for accurate MRI or computed tomography (CT) imaging of convection volume. These methods have been demonstrated in small and large animal models and multiple clinical trials are ongoing to investigate nanoparticle loading with CED.58,59 Nanoparticle formulations of panobinostat, a notoriously hydrophobic and unstable drug, are being investigated in CED clinical trials for DIPG patients.60,61
We once again thank the reviewer for their time.